# Negative Associations between Minority Stressors and Self-Reported Health Status among Sexual Minority Adults Living in Colombia

**DOI:** 10.3390/healthcare12040429

**Published:** 2024-02-07

**Authors:** Paola Roldán, Angela Matijczak, Jacob Goffnett

**Affiliations:** School of Social Work, Virginia Commonwealth University, Richmond, VA 23220, USA; matijczaka@vcu.edu (A.M.); goffnettj@vcu.edu (J.G.)

**Keywords:** sexual orientation, intimate partner violence, vicarious discrimination, self-reported health, Colombia, Latino/a/e/x

## Abstract

Colombia has extensive laws prohibiting discrimination against sexual minority people. However, violence and discrimination toward sexual minorities are still frequent. While a growing body of research shows that sexual minority people experience elevated rates of discrimination and domestic abuse globally, little research has been conducted on these issues affecting sexual minorities in Colombia specifically. Using minority stress theory as a conceptual framework, this paper aims to fill this gap by examining the prevalence of experiencing intimate partner violence (IPV) and witnessed discrimination and the relationship of these stressors to self-reported health among a national sample of sexual minority Colombians. We found that bisexual individuals experienced higher rates of physical and sexual IPV, compared to lesbian and gay individuals. Additionally, sexual minority Colombians who experienced IPV and witnessed discrimination were more likely to report having poorer health, compared to those who had not. We discuss the implications of our findings for future research and clinicians working with sexual minority clients.

## 1. Introduction

Sexual minority Colombians frequently report identity-related victimization and discrimination despite being protected by the country’s broad anti-discrimination laws [1,2]. According to [3], 85% of sexual minority Colombians have experienced violence due to prejudice that includes physical, sexual, patrimonial, economic, and psychological forms. A few empirical studies also demonstrate the prevalence of discrimination experiences in the country [3,4,5]. A study exploring the well-being of sexual minority Colombians [4], found that 25% of participants were discriminated against at work, and 21% received conversion therapy. Scholars attribute these elevated risks to a pervasive stigma toward sexual minority identities and same-sex relationships that is informed by conservative and heterosexist notions embedded in the Latin American culture [6,7,8,9,10].

A burgeoning body of research demonstrates that sexual minority people are at an increased risk of intimate partner violence (IPV) in societies worldwide [11,12,13,14,15,16], but there is a dearth of research on this form of victimization among sexual minority Colombians [5,8,10]. It is critical to investigate the rates and impact of IPV among sexual minority Colombians, as research shows IPV is a prevalent form of victimization in the country [10,17,18,19]. A nationally representative survey showed that around 66% of women aged 13–49 reported encountering some form of IPV in their lifetime [19]. Furthermore, the elevated experience of IPV for Colombian women is associated with a cadre of poor behavioral health outcomes such as increased cortisol levels, psychological distress and smoking, alcohol, and drug use [17,20,21,22]. This association is consistent with the findings of a study exploring domestic violence among same-sex couples in the Colombian Caribbean Region [10].

It is essential to promote optimal health for sexual minority Colombians by conducting further research that explores the population’s experiences of IPV and investigates experiences of discrimination. Additionally, there is an economic imperative underlying such research, as a recent study found that in 2018, the cost of the adverse health outcomes experienced by young Colombian women alone due to physical IPV led to an economic burden of 90.6 million U.S. dollars [17]. The present study explores the prevalence of IPV and witnessed discrimination and the relationship of these stressors to self-reported health among a national sample of sexual minority Colombians. We first overview minority stress theory [23] as an empirical framework for understanding the health of sexual minority Colombians and then detail the relevant minority stress process within a Colombian context. 

### 1.1. Minority Stress Theory

Experiences of victimization and discrimination have been termed distal minority stressors and are enactments of societal and cultural stigma against sexual minority identities and same-sex relationships [23,24,25,26]. Minority stress theory (MST) proposes that elevated rates of distal minority stressors experienced by sexual minority people contribute to the population’s disparately worse health outcomes compared to their heterosexual peers. Support for MST has relied heavily on studies conducted within the U.S. and with predominantly European samples [23,27,28,29]. Few studies have applied or culturally adapted MST to sexual minority individuals living in Latin American countries, such as Colombia. One study guided by MST found higher rates of suicidal ideation among gay men living in Colombia who experienced high levels of internalized homophobia—a proximal minority stressor—or were exposed to sexual abuse [30]. A study conducted by [31] explored whether the minority stress model could be generalized to explain the experiences of sexual minority men living in Brazil. This study’s results supported the use of the minority stress model in non-US samples, showing one form of minority stress, enacted stigma (e.g., harassment, rejection, aggression, violence, or discrimination), to significantly predict depressive symptoms among sexual minority men living in Brazil [31]. However, the authors note that this study was limited in scope due to the inclusion of only a few minority stressors; therefore, further testing of MST in Latin American countries is necessary. 

Studies that have used MST among study samples of Hispanic or Latinx people living in the US have found significant relationships between minority stressors and health outcomes. For example, evidence supports that sexual minority Latino adults who experience unfair treatment (discrimination) have significantly higher levels of psychological distress than those who do not experience unfair treatment [32] and sexual minority Latinx young adults who experience high levels of family rejection have significantly higher levels of depression, substance use, and risky sexual behaviors than those who have no or low levels of family rejection [33]. Three qualitative studies documented the deleterious impact of several forms of structural stigma (e.g., negative religious discourses, intersectional invisibility, and misrepresentation in the media) and interpersonal discrimination (e.g., sexual objectification, gender policing, microinsults) on the health of sexual and gender minority Latinx participants [34,35,36]. It is important to note that even the literature testing MST in samples of sexual minority Latinx individuals living in the US is still emerging. The present study focuses on the relationship between two forms of distal stressors (i.e., IPV and witnessing discrimination) and self-reported health. 

### 1.2. Intimate Partner Violence

Intimate partner violence (IPV) is a form of victimization that occurs in romantic relationships. Exposure to IPV may be conceptualized as a minority stressor, given evidence that sexual minority people may have unique experiences of IPV that specifically relate to their sexual identities (e.g., being outed, homophobia) [37,38]. A few studies have documented the incidence of IPV among sexual minorities in Latin American communities and countries [8,39]. A recent study found that 60.6% of sexual and gender minority respondents in a study in Latin America (mainly in Mexico) reported having encountered some form of IPV at some point in their lives, with psychological aggression being the most frequent [8]. While nearly all studies on IPV in Colombia have focused on heterosexual couples or disregarded sexual orientation, ref. [40] examined the relationship between physical and sexual violence (perpetrated by a family member, partner, acquaintance, or stranger) and health among men who have sex with men (MSM) and transgender women living in Bogotá. The study found experiences of IPV to be associated with binge drinking and drug use [40], consistent with research on the harmful impact of IPV worldwide [41,42,43,44,45].

### 1.3. Witnessed Discrimination

Despite the increasing societal acceptance of sexual minorities, people who self-identify as lesbian, gay, or bisexual continue to confront and witness persistent discrimination in various settings, such as public venues, health facilities, workplaces, and schools [46,47]. Discrimination based on sexual orientation can manifest in direct forms (e.g., being denied jobs or services due to sexual orientation) or indirect, vicarious forms (e.g., witnessing, overhearing, or being aware of others being discriminated against [48]). While most of the research has concentrated on direct discrimination experiences among sexual minorities [48,49], some studies show that vicarious minority stressors also negatively impact the health of sexual minority people [28,48,49]. The present study explores experiences of vicarious discrimination, focusing specifically on witnessing discrimination against sexual minority individuals across various social contexts, including work, school, community, religious, peer, familial, and recreational settings. Research has found that vicarious discrimination toward other sexual minorities is associated with poor mental health outcomes [48,50]. For example, Woodford and colleagues [48] found that sexual minority college students were more likely than heterosexual students to witness hostility (e.g., seeing/hearing someone being verbally threatened, bullied, or intimidated due to sexual orientation), which in turn was associated with higher odds of moderate/high levels of anxiety. Given the prevalence of anti-sexual minority discrimination in Colombia and the paucity of academic research on indirect discrimination faced by sexual minorities [48], investigating the vicarious manifestations of discrimination that sexual minority people face in Colombia is crucial.

### 1.4. Health of Sexual Minority Colombians

The Colombian health system is considered one of the best in Latin America since around 97% of the Colombian population is covered by subsidized or contributory healthcare schemes. Additionally, it boasts one of the lowest out-of-pocket healthcare expenses [51]. However, there are still gaps in effective and quality care, especially among the most marginalized people [3,51,52]. As such, sexual minority Colombians can face barriers to adequate access to health services due to stigmatization and discrimination, mainly evidenced by the lack of clear care criteria that consider this population’s specific needs [3,4,53]. While most sexual minority Colombians perceive their health as good [4,19], this population remains at elevated risk of health-related issues compared to their heterosexual counterparts. Empirical research highlights that the stress associated with sexual identity is pervasive among sexual minority Colombians [30]. Alarmingly, suicide morbidity rates are significantly higher among sexual and gender minority Colombian adults than among cisgender heterosexual individuals, with 56% of sexual and gender minority adults reporting having contemplated suicide at some point in their lives compared to 6.5% of their heterosexual peers [17]. Concerning physical health, sleep disorders are one of the most frequent health problems among sexual minority Colombians [4], and gay and bisexual men are at higher risk of HIV than the general population [4,40,54]. While these studies begin illuminating the extent of health disparities between sexual minorities and heterosexual Colombians, more research is needed to understand social environmental predictors that attenuate the benefits of pervasive access to healthcare to inform prevention and intervention efforts. 

### 1.5. The Current Study

There is a dearth of research on the prevalence of distal minority stressors and their impact on health among sexual minority Colombians [4]. Moreover, no studies have examined IPV and witnessed discrimination among sexual minority Colombians, despite the prevalence of these stressors in other Latin communities and countries, and their association with health outcomes. We address this gap with a national sample of 593 sexual minority Colombian adults. Specifically, we explore differences in the prevalence of experiencing IPV and witnessed discrimination among subgroups of sexual minority Colombians (lesbian/gay vs. bisexual) and examine whether IPV and witnessed discrimination are related to the population’s self-reported health. We hypothesized that having experienced any form of IPV and witnessed discrimination would be associated with poorer self-reported health.

## 2. Materials and Methods

### 2.1. Data and Sample 

We used data from the 2015 Colombian National Demographic and Health Survey (DHS). Data were collected between February 2015 and March 2016 and included a nationally representative sample of 38,718 females aged 13 to 49 years and 35,783 males aged 13 to 59 who voluntarily participated in the survey [19]. Data were gathered in person using a standardized structured questionnaire about violence, fertility, health, and sociodemographic characteristics. For the first time, the 2015 survey version included a set of questions about Colombians’ attitudes and experiences related to the sexual minority community. 

The analytic sample for this study included only participants who self-identified as lesbian, gay, or bisexual (i.e., sexual minority; *n* = 593), given this study’s aims. The sample was majority assigned male at birth (59%) and those who identified as lesbian or gay (56%). Most participants reported being 18–28 years old (58%), having more than a high school education (59%), and never living in a union or married. A total of 91% percent of participants stated having health insurance, and 34% reported having any physical disability. Finally, most participants (51%) reported having good health, with only 11.5% of participants indicating their health was regular or bad (Table 1).

### 2.2. Measures

#### 2.2.1. Dependent Variable

Self-Reported Health Status. Participants’ self-reported health status was measured with the question, “How would you rate your health status in general?”. Participants recorded their responses on a Likert-like scale with options that ranged from bad (1) to excellent (5). Since the number of observations in the category bad was small, observations were recoded to create four categories: bad/regular (1), good (2), very good (3), and excellent (4).

#### 2.2.2. Independent Variables 

Intimate Partner Violence. A modified Conflict Tactics Scale (Straus, 1990 [55]) was used to measure IPV. Participants were asked whether, in their relationship with their current or last partner, they have ever experienced (being victims) (1) physical violence (i.e., partner hit the participant with a fist, kicked her/him, tried to strangle her/him, threatened her/him with a weapon); (2) psychological violence (i.e., partner made decisions without her/him, partner threatened to leave her/him, partner threatened to take children from her/him); or (3) sexual violence (have you ever been physically forced into unwanted sex by the partner?). All response options were dichotomous (yes/no). Sexual violence was measured with one binary item. We recoded observations as having physical violence if the participant answered “yes” to at least one of six questions related to physical violence and recoded to not having experienced physical violence if they indicated “no” to all the questions. The same procedure was followed for psychological violence. 

Witnessed Discrimination. Seven items were used to measure whether participants have ever witnessed a sexual minority person being discriminated against in different social environments, including work, health facilities, school, neighbors, religion, friends, family, and recreational spaces. All responses were dichotomous (yes/no). 

#### 2.2.3. Control Variables

Other sociodemographic variables included are age (a binary variable of age: 18–28 and 29–49), level of education (no education, less than high school, high school, and more than high school), sex assigned at birth (a binary variable: female/male). Dichotomous (yes/no) variables for health insurance and having employment were also included. A binary variable for physical disability was incorporated (i.e., having difficulty hearing voices or sounds, moving around, seeing far/close, talking, communicating, understanding, remembering, making decisions, eating, putting on clothes, taking a shower, relating/interacting with others, and performing daily chores). Observations were recoded as having physical disabilities if participants answered “yes” to at least one of these questions and recoded as not having a physical disability if they indicated “no” to all questions. Finally, household economic status was measured by the wealth index (WI) the DHS created. The WI measures a household’s living standard, representing the quintiles of the national Colombian wealth distribution (poorest, poorer, middle, richer, and richest). In the analyses, we recoded the lowest two categories (poorest and poor) as poor and the highest two categories (rich and richest) as rich.

#### 2.2.4. Analytic Approach 

We first conducted a bivariate analysis to test whether IPV outcomes and witnessed discrimination varied by sexual orientation (i.e., bisexual vs. lesbian or gay [LG] people) using chi-square. Next, we estimated the association between self-reported health status and independent variables using multivariate ordered logistic regression.

Since the parallel assumption of the ordered logit model was violated, we estimated the partial proportional odds model, a variant of the generalized ordered logit model in which coefficients of the regressions are constrained to be the same only for the variables that did not violate the parallel assumption (Williams, 2006 [56]). One advantage of this model is its parsimoniousness since it reduces the number of estimated coefficients compared to unconstrained or multinomial regression models.

To gauge the combined effect of sexual orientation and experiencing physical violence from a partner, we included an interaction term of being bisexual and experiencing intimate partner physical violence. Then, we added the coefficients of being bisexual and the coefficient of the interaction term to conduct a linear test hypothesis of these combined coefficients for each model (regular/bad, good, and very good), using Bonferroni-adjusted *p*-values. 

All analyses were conducted in Stata 17. Sample weights and heteroskedasticity robust standard errors were used.

## 3. Results

### 3.1. Bivariate

The results from the analysis show that victimization experiences differed significantly between bisexual and LG participants. Physical and sexual violence were more prevalent among people who self-identified as bisexual (40.4%, *p* < 0.01; 8.6%, *p* < 0.01, respectively) than LG participants (25.6%; 2.2%, respectively). Psychological violence was prevalent among all participants: 65% of participants, regardless of their sexual orientation, reported having experienced psychological violence from their partners. The first section of Table 2 shows the percentage of participants who experienced IPV, according to their sexual orientation.

As shown in the second section of Table 2, witnessed discrimination based on sexual orientation is frequent in Colombian society. As reported by respondents, almost 60% of participants, regardless of their sexual orientation, had witnessed a sexual minority person being discriminated against at school (59.1%), by friends (60%), and by family (53%). The analysis also showed significant differences between bisexuals and LG participants. A higher percent of LG participants, relative to bisexual participants, reported ever witnessing a sexual minority person being discriminated against by neighbors (61.9% vs. 53.8%, *p* < 0.01), at work (48.6% vs. 43.1%, *p* < 0.01), and in religious groups (39.6% vs. 30.5%, *p* < 0.01) and recreational places (30.8% vs. 23.7%, *p* < 0.01).

There were no significant differences between LG and bisexual people regarding self-reported health. Most participants stated they have good health (51%), very good (10.8%), or excellent health (22.7%), while only 15.5% of participants reported bad/regular health.

### 3.2. Multivariate Analyses

Overall, the results from the partial proportional odds model (Table 3) indicate that being discriminated against at work and having any physical disability were associated with a lower likelihood of reporting excellent health status. Specifically, sexual minority participants who ever witnessed a sexual minority person being discriminated against at work have a lower likelihood of reporting excellent health status than those who did not witness this type of discrimination (coef. = −1.0, *p* < 0.05). Similarly, sexual minority people who reported having any physical disability hold a lower likelihood of reporting excellent health status than people who did not report having physical disabilities (coef. = −1.8, *p* < 0.01). In contrast, LGB people classified as middle income in the DHS wealth index hold a higher likelihood of reporting excellent health status than people classified as poor (coef. = 0.89, *p* < 0.05). 

Our model also shows that identifying as bisexual and experiencing physical violence from a partner is associated with a lower likelihood of reporting excellent health status. In the regular/bad health status model, bisexual people, regardless of whether or not they have been exposed to physical violence from a partner, have a higher likelihood of reporting excellent health status rather than a regular/bad one (coef. = 2.16, *p* < 0.01). Furthermore, identifying as bisexual and experiencing intimate partner physical violence is associated with a lower likelihood of reporting excellent health rather than a regular/bad health status (coef. = −3.59, *p* < 0.01); however, the linear test hypothesis of the combined coefficients of being bisexual and having been exposed to intimate partner physical violence is not rejected to be zero (combined coef. = −1.43, chi2 = 4.81, *p* = 0.08). This means that the effects cancel each other; therefore, the net effect of bisexuals being exposed to intimate partner physical violence is not significant in this model.

In the good health status model, the coefficient of the bisexual variable was negative but not significant (coef. = −1.46, *p* = 0.09), and the coefficient of the interaction between bisexuals and bisexuals experiencing physical violence was negative but not significant (coef. = −0.42, *p* = 0.07); however, the linear test hypothesis of the combined coefficients of being bisexual and having been exposed to physical violence from a partner is rejected to be zero (combined coef. = −2.2, chi2 = 7.17, *p* < 0.05). This means that bisexuals who are exposed to physical violence from a partner are associated with a lower likelihood of reporting excellent health status rather than good health status. 

In the very good health status model, bisexual people, regardless of whether having been exposed or not to intimate partner physical violence, have a higher probability of reporting excellent health status rather than very good (coef. = 3.46, *p* < 0.01) and bisexuals who experienced physical violence had a lower probability of reporting excellent health status (coef. = −5.47, *p* < 0.01); the linear test hypothesis of the combined coefficients of being bisexual and experiencing physical violence is rejected to be zero (combined coef. = −2.00; chi2 = 7.80, *p* < 0.05). This means that experiencing physical violence is associated with a lower probability of reporting excellent health status by bisexuals.

## 4. Discussion

IPV is a prominent public health concern for sexual minority people in many countries [11,12,13,14,15,16], but research has yet to identify the scope of risk in Colombia. We addressed this gap by examining the prevalence of physical, psychological, and sexual IPV among a national sample of sexual minority Colombians while also evaluating differences between sexual minority subgroups (bisexual vs. lesbian/gay). In addition, we assessed the prevalence of another distal minority stressor, witnessed discrimination, which has been relatively understudied compared to direct experiences of discrimination [48,49], and examined the relationship between both minority stressors and self-reported health. Our findings are congruent with past research on the experience and impact of these stressors across countries and within other Latin communities and provide vital information on factors that place sexual minority Colombians at risk of poor health despite having widespread access to healthcare. 

Our study is the first to examine multiple forms of IPV among a national sample of sexual minority Colombians. The results show a higher risk of physical and sexual violence for bisexuals compared to lesbian or gay Colombians. This finding is consistent with the broader literature on IPV that shows bisexual people are at higher risk than people identifying with other sexual orientations, including heterosexuals [16,57,58,59,60,61]. Binegativity, which refers to the negative stereotypes and beliefs associated with bisexuality (e.g., promiscuity, attention-seeking, and perceived instability or confusion in their sexual orientation), is widespread in both heterosexual and sexual minority communities [16,61] and may heighten the risk of IPV and consequent health problems for bisexual people [16,57,58,59,60]. Scholars have argued that negative stereotypes, such as promiscuity, may lead to partner jealousy, which has been connected to a higher risk of IPV among heterosexual people [57]. People who are not bisexual often consider that bisexual partners are just going through a temporary stage and will inevitably leave their relationships when they realize they are gay and not bisexual [16]. This perception can contribute to stereotypes that bisexual individuals are unpredictable in their relationships [16,57,62], potentially causing insecurity among their partners and increasing the risk of IPV. Nevertheless, despite these prevalent beliefs, it is worth noting that stereotypes about bisexuality have found little support in empirical research [16,63,64].

We also observed a high prevalence of psychological IPV for all sexual minority participants, a finding consistent with past research in Colombia [5], Chile [65], and broader IVP studies’ findings showing that psychological IVP is far more prevalent than physical or sexual violence for LGBT people [66,67]. Ref. [66] examined the occurrence of perpetration and victimization in several types of IPV among lesbian women in Turkey, with the highest prevalence being psychological violence perpetration (66.4%) and victimization (63.1%). However, research has found differences in the prevalence of psychological partner violence across sexual orientations when examining behaviors with more granularity. Ref. [68] found that lesbians, compared to bisexual women, had a greater prevalence of victimization related to control behaviors and verbal aggression. 

Finally, we show that the experiences of physical IPV and witnessing discrimination toward another sexual minority person may decrease the likelihood of reporting excellent health for participants. Numerous studies show that victims of IPV are at higher risk of reporting negative physical and mental health outcomes such as anxiety, depression, low self-esteem, gastrointestinal disorders, substance abuse, sexually transmitted diseases, and unwanted pregnancy [15,67,69,70,71,72,73]. Our study contributes to this literature by demonstrating a connection between physical IPV and self-reported poorer health for sexual minority Colombians. We also demonstrate that witnessing a sexual minority person being discriminated against decreases the likelihood of reporting excellent health for participants. Although the impact of vicarious discrimination on health outcomes is understudied [50], our result is consistent with the research that exists on the topic, which shows witnessed discrimination to be associated with anxiety, depression, suicidal ideation, and alcohol abuse [48,50].

## 5. Limitations

Using a national sample of Colombians, we have produced several novel findings imperative to increasing the knowledge base on the experiences of minority stressors among sexual minority Colombians and developing methods for preventing and intervening in the impact these stressors have on health outcomes. Yet, our study is not without limitations. The cross-sectional design implies that causality cannot be determined but does provide a strong foundation for future research. Furthermore, the use of self-reported health is broad and subjective. While researchers have demonstrated satisfactory overlap between self-reported health and more objective assessments (e.g., laboratory blood tests and doctor’s diagnosis), scholars have cautioned that self-reported health may better capture certain diseases (e.g., cardiovascular disease, mental health problems) and may be affected by a country’s public health and medical systems [74,75]. Finally, the modified Conflict Tactics Scale is one of the most commonly used measures of IPV, yet it was created and validated with heterosexual populations [76]. Thus, it may not be the most appropriate measure of IVP among sexual minorities due to their unique experiences.

## 6. Conclusions

The novel findings from the current study offer several directions for future research as well as considerations for prevention and intervention work with sexual minority Colombians. While we did not assess the role of gender in experiences of IPV within our sample, several studies find that sexual minorities assigned female at birth and bisexual women are at higher risk of experiencing IPV compared to other sexual identities, including heterosexuals [11,57,61,73]. Recent research outside of Colombia also shows a high prevalence of IPV among transgender and gender-diverse folks [77], including significantly higher rates for the population compared to cisgender people [78]. Additionally, this research shows that dating violence, in part, accounts for the statistically significant relationship between being transgender or gender diverse and suicidal ideation [79]. Thus, future research should build from our findings by examining the influence of gender on IPV rates among sexual and gender minority Colombians and the impact these experiences have on different health outcomes. Future research should continue to investigate witnessed discrimination as a prevalent experience among sexual minority Colombians that impacts their health outcomes by understanding constructs that facilitate this relationship. Scholars propose that cognitive and affective processes, such as group-based emotions, fear of direct discrimination, and perceived social exclusion, may connect witnessed discrimination to health outcomes, particularly suicidality [50]. 

## 7. Implications

Healthcare clinicians in Colombia should be mindful of the prevalence of IPV and witnessed discrimination when working with sexual minority clients. It is important to assess whether these experiences are relevant to a client and to identify any influence on health outcomes. While we only examined self-reported health, several studies have shown LGB-affirmative treatment modalities to reduce maladaptive behavioral health symptoms (e.g., alcohol misuse, depression, anxiety, and PTSD; [80,81,82]) that may be effective when adapted to the culturally relevant experiences of sexual minority Colombians. Bystander intervention may also prove to help prevent LGB mistreatment of sexual minorities. The bystander perspective prioritizes education and training programs to reduce mistreatment by helping concerned individuals develop awareness, personal responsibility, and empowerment to intervene when witnessing violence or harassment [83,84,85]. Bystander training and their evaluations have shown the effectiveness of observers acting to reduce mistreatment incidents based on sexual orientation and promote inclusive environments in schools, colleges, and workplaces [83,84,85,86,87]. However, scholars observe that men may be less willing to intervene than women when witnessing mistreatment based on sexual orientation and this may be particularly true in a patriarchal Colombian society that still cultivates masculine norms that are homophobic [88]. Therefore, some men may be involved in homophobic behavior or approve of it when observed. At the same time, others may wish to counter homophobic behavior but may fear social judgment for challenging behaviors related to masculinity norms [89]. In this sense, bystander training should be tailored to issues of masculinity and may be most effective when aimed at men and boys rather than women and girls.

## Figures and Tables

**Table 1 healthcare-12-00429-t001:** Sample characteristics (*n* = 593).

Sociodemographic Characteristics	Full Sample (%)	*n*
Sex assigned at birth (men)	58.8	348
Sexual Orientation (Gay/Lesbian)	56.4	334
Education		
High school diploma	34.4	243
More than high school	59.4	301
Age, years		
18–28	58.1	360
Marital Status		
Never in union	64.7	384
Living union/married	18.1	197
Divorced/widowed	17.2	102
Physical Disability (Yes)	34.0	200
Health Insurance (Yes)	90.9	539
Currently Employed (Yes)	78.4	466
Houshold Characteristics		
Wealth Index		
Poor	23.9	124
Middle	20.88	124
Rich	55.18	327
Independent Variables		
Intimate Partner Violence (Yes)		
Physical violence	29.2	68
Psychological violence	60.4	140
Sexual violence	3.7	9
Witnessed Discrimination (Yes)		
By neighbors	53.2	316
In fun places	27.6	164
At work	31.8	189
By religious groups	34.7	206
In health facilities	7.2	43
In school	56.6	336
By friends	55.9	332
By family	50.7	301
Dependent Variable		
Health Status		
Excellent	24.3	144
Very good	10.8	64
Good	53.5	317
Regular/bad	11.5	68

**Table 2 healthcare-12-00429-t002:** Experiences of victimization and discrimination by sexual orientation.

		Sexual Orientation		
	Full Sample (%)	Bisexual Person (%)	Lesbian/Gay Person (%)	
Victimization (Yes)				
Physical violence	32.99	40.43	25.6	***
Sexual violence	5.40	8.6	2.2	***
Psychological violence	64.94	65.96	64.2	
Discrimination (Yes)				
Neighbors	57.9	53.8	61.9	**
Fun places	27.2	23.7	30.8	**
Religious groups	35.1	30.5	39.6	**
At work	45.9	43.1	48.6	**
At school	59.1	58.4	59.8	
By family	52.8	51.2	54.4	
Health facilities	6.9	7.9	5.7	

*** *p* < 0.01, ** *p* < 0.05.

**Table 3 healthcare-12-00429-t003:** Factors associated with self-reported health status.

Factors Associated with Health Status	b	SE	
Variables not violating the PL assumption			
Sociodemographic Characteristics			
Physical disabilities (ref. no)	−1.80	0.38	***
Wealth index (ref. poor)			
Middle income	0.89	0.45	**
Discrimination			
At work	−1.00	0.41	**
Variables violating the PL assumption			
Model for regular/bad Health Status			
Sexual Orientation (ref. LG)			
Bisexual	2.16	0.79	***
Victimization			
Physical Violence × Sexual Orientation (ref. Lesbian/Gay)			
Yes × Bisexual	−3.59	1.04	***
Model for good Health Status			
Sexual Orientation (ref. LG)			
Bisexual	−1.46	0.86	
Victimization			
Physical Violence × Sexual Orientation (ref. Lesbian/Gay)			
Yes × Bisexual	−0.42	1.12	
Model for very good Health Status			
Sexual Orientation (ref. LG)			
Bisexual	3.46	1.1	***
Victimization			
Physical Violence × Sexual Orientation (ref. Lesbian/Gay)			
Yes × Bisexual	−5.47	1.37	***
Observations 593			

SE: robust standard errors, *** *p* < 0.01, ** *p* < 0.050.

## Data Availability

Publicly available datasets were analyzed in this study. These data can be found here: https://dhsprogram.com/ (accessed on 15 June 2022).

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
