# Peer review of "Negative Associations between Minority Stressors and Self-Reported Health Status among Sexual Minority Adults Living in Colombia"

_healthcare, 2024, doi:10.3390/healthcare12040429_

Round 1

Reviewer 1 Report

Comments and Suggestions for Authors

Line 121: It would be helpful if the authors briefly detailed the nature of the partial mediation that occurred.

Line 164: use “Data were…”

Line 175: For maximal clarity, the authors should report the proportion that indicated their health was good, very good or excellent. When I first read the manuscript, I assumed that, as 51% reported their health was good, 49% felt their health was bad. 

Line 182: What proportion of respondents indicated that their health was “bad”?

Line 196: I assume the same coding procedure was used for sexual violence.

Line 214: Was wealth distribution recoded (see Table 1)?

Table 1 reveals that 55.8% of participants were classified as “rich.”  Is this proportion surprising?

Line 226: For bisexual participants, can the authors determine whether the physical and sexual violence were more likely to occur with same-sex versus other-sex partners?1

Comments on the Quality of English Language

The English language is used appropriately. 

Author Response

Line 121: It would be helpful if the authors briefly detailed the nature of the partial mediation that occurred 

We appreciate the recommendation and have added language that describes the mediation. 

Line 164: use “Data were…”

Thank you for noticing this mistake. We made the correction. 

Line 175: For maximal clarity, the authors should report the proportion that indicated their health was good, very good or excellent. When I first read the manuscript, I assumed that, as 51% reported their health was good, 49% felt their health was bad.  

We have specified how many participants endorsed having regular or bad health in the text of the manuscript:  “with only 11.5% of participants indicating their health was regular or bad.”

Line 182: What proportion of respondents indicated that their health was “bad”? 

Only 1% of the participants reported having bad health

Line 196: I assume the same coding procedure was used for sexual violence. 

Sexual violence was measured using one dichotomous variable. We have specified this in the manuscript. 

Line 214: Was wealth distribution recoded (see Table 1)? 

Yes, poorer and poor were recoded as ‘poor’ and richest and rich as ‘rich.’ We have added this description to the manuscript.

Table 1 reveals that 55.8% of participants were classified as “rich.”  Is this proportion surprising? 

It is not necessarily surprising since nearly 60% of our sample of LGB Colombians have more than a high school education. 

Line 226: For bisexual participants, can the authors determine whether the physical and sexual violence were more likely to occur with same-sex versus other-sex partners? 

Unfortunately, it is not possible to determine whether physical and sexual violence were more likely to occur with same-sex versus other-sex partners from this data.

Reviewer 2 Report

Comments and Suggestions for Authors

My genareal evaluaion is very positive. It is very interesting and inovative article. Anyway, I desagree with the concep of Minority Stress Theory (MST). You should consider that heterosexual people are also a minority group but the neoliberalism needs still to reinforce the hegemony of this model to mantein opression of women. Who has invented the concept of minority? Who needs this concept? A critical view is necesary to understand better tje problem of the stress.. Intimate seuxal viiolence should be explain better because they are sexual violence issues. Finally the selection of sample need to clear the criterium for the selection, "convinience" "based on criterium" or any other. I am not sure if you have included MST  after using it the frist time in the text. The problems they have are the same of heterosexual people as Badinter, E, expained in XY La identidad masculina. Results and conclusion are good.

Author Response

The 2015 Colombian Demography Health Survey is a national, representative, and population-based survey that used stratified, two-stage cluster sampling to select households invited to participate in the survey. 

We appreciate the reviewers comments. MST is a widely used and empirically supported framework for understanding disparities in health outcomes for LGB+ people by examining the occurrence and impact of stressors unique to or disparately higher among the population. These stressors are driven by a pervasive stigma against non-heterosexual identities (e.g., those who are lesiban, gay, and bisexual) that exist in many societies around the globe. Indeed, this research consistently shows that –even in societies with increased inclusivity of LGB people – disparately high rates of discrimination and victimization exist leading to poorer health (see Frost, D. M., & Meyer, I. H. (2023). Minority stress theory: Application, critique, and continued relevance. Current Opinion in Psychology, 101579). However, this framework has not been widely used in Colombia, so the aim of our study was to examine the relevance of MST among this population. We feel the background, methodology, and discussion are consistent with this application. 

Reviewer 3 Report

Comments and Suggestions for Authors

Using the 2015 Colombian National Demographic and Health Survey 161 (DHS), this study examined the linkages between sexual minority stressors and self-reported health status in Columbia. My comments are included below.

Strengths: (1) Good use of the minority stressor theory that provides conceptual linkages between sexual minority experiences of stress and self-reported health status. (2) The authors were able to identify two major mechanisms that are significantly and positively associated with poorer health status, namely, experiencing psychological violence and discrimination at work net of several sociodemographic characteristics.

Weaknesses: (1) Table 1 should include descriptive statistics for all the dependent, independent, and control variables. (2) The authors should try to develop a count or index variable that encompasses all the IPV measures and include it in their multivariate regression analysis (ordered logit models). Make sure to report the reliability coefficient and the regression results (3) By the same token, the authors should try to develop an index variable of discrimination (report the reliability coefficient) by combining all the discrimination measures and include this variable in their multivariate regression models. Report the results. (4) After running the ordered logit models, did the authors check the proportional odds assumption? If violated, please use the generalized order regression models developed by Richard Williams (see “Generalized ordered logit/partial proportional odds models for ordinal dependent variables” The Stata Journal (2006) 6, Number 1, pp. 58–82).

Overall suggestion: Though this study provides some empirical insights into the linkages between sexual minorities’ experiences of stressors derived primarily from IPV and discrimination and self-reported health, I wonder if these findings are indeed unique to this subpopulation. To find an answer, I would suggest that authors conduct the same analyses for their heterosexual subsample and compare and contrast differences and similarities, which will provide a stronger test for the minority stressor theory or model in the context of Columbia. However, I defer the decision to the editors.

Comments on the Quality of English Language

The Quality of the English Language is good. Please line-edit the manuscript to check spelling and readability.

Author Response

Strengths: (1) Good use of the minority stressor theory that provides conceptual linkages between sexual minority experiences of stress and self-reported health status. (2) The authors were able to identify two major mechanisms that are significantly and positively associated with poorer health status, namely, experiencing psychological violence and discrimination at work net of several sociodemographic characteristics.

We appreciate the reviewer highlighting the strengths of our manuscript. 

Weaknesses: (1) Table 1 should include descriptive statistics for all the dependent, independent, and control variables. 

We have updated Table 1 to include all variables included in the analyses.

The authors should try to develop a count or index variable that encompasses all the IPV measures and include it in their multivariate regression analysis (ordered logit models). Make sure to report the reliability coefficient and the regression results 

The goal of the manuscript was to examine whether ever experiencing different forms of IPV are related to self-reported health outcomes. An index variable would examine if more experiences of IPV or greater exposure to different forms of IPV relate to self-reported health, which was not the aim of this study. We have clarified the aims of our study in the manuscript. Additionally, we do not feel we have the appropriate items to assess this since participants were asked if this ever happened in the past 12 months, but not how many times it has happened. We believe this creates an issue where we are only measuring the variety of IPV experiences a participant was exposed to, but this does not equate to the frequency of IPV exposure (as someone may have been exposed to one form of IPV several times). Further, index measures of IPV treat different forms of IPV (e.g., physical, sexual) as equivalent, which is a serious limitation. Based on support from previous work about the limitations of an index variable to measure IPV and trauma exposure (Rasmussen et al., 2020; Williams et al., 2020), we have opted not to include an index variable for IPV or discrimination in our models at this time. We are amenable to including these variables if the reviewers and/or editors deem it necessary. 

Rasmussen, A., Romero, S., Leon, M., Verkuilen, J., Morales, P., Martinez‐Maganalles, S., & García‐Sosa, I. (2020). Measuring trauma exposure: Count versus variety of potentially traumatic events in a binational sample. Journal of Traumatic Stress, 33(6), 973-983. https://doi.org/10.1002/jts.22563 

Williams, J. R., Burton, C. W., Anderson, J. C., & Gonzalez-Guarda, R. M. (2020). Scoring interpersonal violence measures: Methodological considerations. Nursing research, 69(6), 466. https://doi.org/10.1097%2FNNR.0000000000000461 

 By the same token, the authors should try to develop an index variable of discrimination (report the reliability coefficient) by combining all the discrimination measures and include this variable in their multivariate regression models. Report the results 

Similarly to our response to the comment above, it was not within the aims of this paper to assess the relationship between cumulative exposure to discrimination and self-reported health status. We also feel there are limitations to using an index variable, as participants were not asked how many times they were exposed to each form of discrimination. Based on recommendations from previous work that suggests assessing variety of potentially traumatic experiences is not a good proxy for assessing frequency of exposure (Rasmussen et al., 2020), we have opted not to include an index variable at this time. However, we are amenable to including this variable if the reviewers and/or editors deem it necessary. 

Rasmussen, A., Romero, S., Leon, M., Verkuilen, J., Morales, P., Martinez‐Maganalles, S., & García‐Sosa, I. (2020). Measuring trauma exposure: Count versus variety of potentially traumatic events in a binational sample. Journal of Traumatic Stress, 33(6), 973-983. https://doi.org/10.1002/jts.22563 

 (4) After running the ordered logit models, did the authors check the proportional odds assumption? If violated, please use the generalized order regression models developed by Richard Williams (see “Generalized ordered logit/partial proportional odds models for ordinal dependent variables” The Stata Journal (2006) 6, Number 1, pp. 58–82). 

Thank you for this comment and for suggesting this interesting and pertinent document.  We checked the proportional odds assumption and it was violated, so we estimated the partial proportional odds model. We addressed this comment in the analytic approach section of the document and results of the partial proportional odds model are shown in table 3. 

Overall suggestion: Though this study provides some empirical insights into the linkages between sexual minorities’ experiences of stressors derived primarily from IPV and discrimination and self-reported health, I wonder if these findings are indeed unique to this subpopulation. To find an answer, I would suggest that authors conduct the same analyses for their heterosexual subsample and compare and contrast differences and similarities, which will provide a stronger test for the minority stressor theory or model in the context of Columbia. However, I defer the decision to the editors. 

We appreciate the suggestion but our goal was to examine minority stress processes among LGB+ Colombians and to look at differences within these identity categories. We believe that comparing these experiences to those of heterosexuals alters the aim of the current study. 

Round 2

Reviewer 3 Report

Comments and Suggestions for Authors

The authors were by and large responsive. However, it seems that they wanted to avoid conducting more data analysis, which I do understand. For example, I would assume that in Columbia if sexual minorities experienced multiple forms of IPV would be more likely to report negative health outcomes, whereas analysis for each type of IPV could be problematic, due to, e.g., skewed distributions (see sexual IPV: only 3.7%). Even if this type of analysis produced no significant results, it would be nice to see them in the revision report, which does not have to be included in the paper.

Moreover, I suggested running the same analysis for the heterosexual respondents included in the survey to check the theoretical framework. But the authors declined. That is OK. Sometimes, sensitive analysis can help provide stronger evidence. 

It is great to see that the authors were able to check the proportional odds assumptions for their ordered logit models. the results seem to be robust. thank you. 
